# Electrospray-Assisted Fabrication of Dextran–Whey Protein Isolation Microcapsules for the Encapsulation of Selenium-Enriched Peptide

**DOI:** 10.3390/foods12051008

**Published:** 2023-02-27

**Authors:** Jiangling He, Zhenyu Wang, Lingfeng Wei, Yuanyuan Ye, Zia-ud Din, Jiaojiao Zhou, Xin Cong, Shuiyuan Cheng, Jie Cai

**Affiliations:** 1National R&D Center for Se-Rich Agricultural Products Processing, Hubei Engineering Research Center for Deep Processing of Green Se-Rich Agricultural Products, School of Modern Industry for Selenium Science and Engineering, Wuhan Polytechnic University, Wuhan 430023, China; 2Key Laboratory for Deep Processing of Major Grain and Oil, Ministry of Education, Hubei Key Laboratory for Processing and Transformation of Agricultural Products, Wuhan Polytechnic University, Wuhan 430023, China; 3Department of Food Science and Nutrition, Women University Swabi, Swabi 23430, Khyber Pakhtunkhawa, Pakistan

**Keywords:** selenium-enriched peptide, microcapsule, electrospray, release performance, cytotoxicity evaluation

## Abstract

Selenium-enriched peptide (SP, selenopeptide) is an excellent organic selenium supplement that has attracted increasing attention due to its superior physiological effects. In this study, dextran–whey protein isolation–SP (DX-WPI-SP) microcapsules were fabricated via high-voltage electrospraying technology. The results of preparation process optimization showed that the optimized preparation process parameters were 6% DX (*w*/*v*), feeding rate *Q* = 1 mL/h, voltage *U* = 15 kV, and receiving distance *H* = 15 cm. When the content of WPI (*w*/*v*) was 4–8%, the average diameter of the as-prepared microcapsules was no more than 45 μm, and the loading rate for SP ranged from ~46% to ~37%. The DX-WPI-SP microcapsules displayed excellent antioxidant capacity. The thermal stability of the microencapsulated SP was improved, which was attributed to the protective effects of the wall materials for SP. The release performance was investigated to disclose the sustained-release capacity of the carrier under different pH values and an in-vitro-simulated digestion environment. The digested microcapsule solution showed negligible influence on the cellular cytotoxicity of Caco-2 cells. Overall, our work provides a facile strategy of electrospraying microcapsules for the functional encapsulation of SP and witnesses a broad prospect that the DX-WPI-SP microcapsules can exhibit great potential in the food processing field.

## 1. Introduction

Selenium (Se) is an indispensable micronutrient for human health. In nature, Se is mainly present in two forms, which are inorganic selenium (selenite, sodium selenite) and organic selenium (seleno amino acids, selenium polysaccharides, selenoproteins, etc.) [1]. Organic selenium has the superiorities of lower toxicity and higher bioavailability compared to inorganic selenium. Organic selenium is also much safer and can be used as a nutritional supplement for selenium-deficient people. Previous studies have claimed that the intake of super-nutritional selenium can fight cancer [2]. Therefore, people in selenium-deficient areas can select selenium-rich foods as dietary additives to keep healthy. Plant-derived selenium-enriched peptide (SP, selenopeptide) has a high biological activity, and this superiority makes it an important source of selenium supplements. The intake of SP can endow the human body with many biological effects, such as antioxidant agents [3], fighting cancer [4], fighting inflammation [5], and immune regulation [6]. However, SP is unstable, easily denatured, and inactivated under the stimulation of oxygen, heat, alkali, etc. It is necessary to overcome these weaknesses to cater to food and other applications.

Microencapsulation has been utilized to protect functional ingredients from harsh factors [7]. The unstable matter is encapsulated within polymers to form a composite to protect the matter from the external environment. Microencapsulation technology can overcome these problems and promote applications in the food field, such as the encapsulation of fats/oil [8,9], vitamins [10], minerals [11], and biologically active substances [12,13,14]. The methods to construct microcapsules are mainly divided into chemical methods (in-situ polymerization, interfacial polymerization, complex coacervation), physical methods (air suspension method, spray drying method), and physical–chemical methods (sol-gel encapsulation method, supercritical CO_2_-assisted method) [15]. Ko et al. prepared sulforaphane microcapsules via freeze-drying with poly(d,l-lactide-co-glycolide) as the wall material for osteoarthritis treatment [16], but the residual toxic solvents can seriously impede the applications.

Electrospraying technology can be used to prepare microparticles or nanoparticles via an electrohydrodynamic jet [17]. In recent years, it has been regarded as a direct and common method to fabricate nanoparticles and microparticles [18,19,20]. Compared with traditional technologies, electrospraying technology has the advantages of high efficiency and high yield. The electrospraying technology relies on a physical electric field to fabricate microcapsules without the use of heat, where the activity of functional ingredients can be well maintained. Thus, this technology is an excellent tool for encapsulating heat-sensitive substances. The as-prepared particles via electrospraying technology have uniform and controllable size distribution, good repeatability, and high production efficiency. Therefore, more attention has been paid to the research of electrospraying for the embedding and delivery of active substances [21,22,23]. Electrospraying technology has been used for the biopolymer encapsulation of functional food extracts to enhance their stability [24]. Costamagna et al. reported that a Chañar extract, rich in polyphenols, was encapsulated into the zein shell by electrospraying, where the stability of the extract was significantly improved [25]. Similarly, Mahalakshmi et al. prepared microcapsules via electrospraying technology to effectively embed *β*-carotene into zein to enhance its stability against surrounding environments [26].

Electrospraying is a gentle treatment method, which is similar to electrospinning technology, and an electrospraying device generally consists of three parts: (1) an injection module (syringe and syringe pump); (2) a high-voltage generator connected to a syringe needle for generating high voltage; (3) a receiving module (titanium alloy plate, rotating titanium alloy receiving rod or receiving solution). For the electrospraying process, it is essential to use a polymer with good solubility. Whey protein isolation (WPI) is a kind of excellent water-soluble and food-grade material with high nutrition. In addition, WPI has the potential to be used as functional encapsulation and a controlled-release delivery system due to its easily functional property and excellent biocompatibility [27]. However, excessive dissolution is a major challenge for some bioactive compounds. Dextran (DX) is an electroneutral and water-soluble polysaccharide with numerous physiological effects. Reported work has demonstrated that the introduction of DX can delay the dissolution rate of protein [28]. The interaction between proteins and polysaccharides can improve the structure, stability, encapsulation, and other specific functional properties [29]. Moreover, there are rare reports on interactions between proteins and non-charged polysaccharides. The interaction mechanism between proteins and uncharged polysaccharides is still not clear, especially regarding the ternary composite system (DX, WPI, and SP). Therefore, the study of the physicochemical properties and controlled-release properties of protein–peptide–polysaccharide ternary composite microcapsules is significant for exploring the potential prospects of SP in food fields.

Herein, electrospraying technology is employed to fabricate ternary composite microcapsules composed of DX and WPI as the wall material and SP as the core material. The utilized parameters to prepare DX-WPI-SP microcapsules are as follows: 6% DX, feeding rate (*Q*) 1.0 mL/h, voltage (*U*) 15 kV, and receiving distance (*H*) 15 cm. The physicochemical properties of these ternary (DX-WPI-SP) microcapsules are investigated, and their antioxidant capacities are also analyzed. Subsequently, the preparation process is optimized to obtain DX-WPI-SP microcapsules with a higher loading rate and a smaller and more uniform particle size distribution. The release properties of the microcapsules are evaluated through the fitting of four common release models (zero order, first order, Higuchi, and Korsmeyer–Peppas) under different pH value environments (pH 2.0, pH 7.0, and pH 12.0). Furthermore, the release behavior of the DX-WPI-SP microcapsules in the simulated human digestive fluid is explored, and the biotoxicity of post-digestive ternary microcapsules is assessed. The DX-WPI-SP microcapsules exhibit great gastrointestinal dissolution characteristics and controllable release properties for SP. This work focuses on the construction of an efficient vehicle to deliver and release SP and then excavates the potential of SP as a novel selenium supplement in the food industry.

## 2. Materials and Methods

### 2.1. Materials and Reagents

Materials and reagents are recorded in the Appendix A.

### 2.2. Characterizations

The main characterizations and instruments are presented in the Appendix A.

### 2.3. Preparation of DX-WPI-SP Microcapsules

High-voltage electrospraying technology was used to prepare the microcapsules. The high-voltage electrospraying device was composed of a DW-N503-1ACDF0 high-voltage DC power device, an automatic injection pump, and a receiver. The preparation process referred to the reported method of Paximada et al. with some changes [30]. Anhydrous ethanol was added to the receiver to disperse the microcapsules and to remove the water within the microcapsules. Specifically, the electrostatic spray solution was dissolved in 10 mL of DI water and transferred to a 10 mL syringe. Subsequently, this syringe was fixed on the groove of the automatic syringe pump and connected with a needle through a Teflon catheter. The needle was connected to the positive pole of the high-voltage power supply, the receiver was grounded, and both of them were placed vertically.

### 2.4. Antioxidant Activity Evaluation

#### 2.4.1. DPPH Radical Scavenging Assay

DPPH scavenging capacity was based on the previous report by Zhang et al., with some changes [31]. Specifically, different microcapsule solutions were mixed with 1 mL of 0.2 mM DPPH, and then the mixed solution was incubated in the dark for 30 min. The absorbance value of the mixture was recorded by the UV–vis spectrophotometer at 517 nm. The DPPH free radical scavenging rate calculation formulation is presented in the Appendix A.

#### 2.4.2. ABTS Radical Scavenging Ability Assay

The free radical ABTS scavenging ability of the microcapsules was evaluated following the work reported by Sridhar et al., with some modifications [32]. Specifically, the ABTS radical stock solution was produced by mixing ABTS aqueous solution (7.0 mM) with a 4.80 mM aqueous solution of potassium persulfate in equal volumes. After that, the mixture was left to react for 16 h in the dark. The ABTS stock solution was diluted to an optical density (OD) at 0.700 ± 0.001 at the optical length of 1 cm, which was defined as the ABTS work solution. The sample was poured into a 10 mL centrifuge tube, and then fresh ABTS work solution (3.8 mL) was added. The mixed solution was shaken and stood in the dark for 6 min. The absorbance value of the resultant supernatant was recorded at 734 nm. The ABTS free radical scavenging rate calculation formulation is presented in the Appendix A.

### 2.5. Stability of DX-WPI-SP Microcapsules against pH Changes

The stability of active substances can be improved after being encapsulated in microcapsules under various pH changes [33]. Herein, the pH stability of DX-WPI-SP microcapsules was evaluated by investigating the release rate of SP under different pH values and plotting the cumulative release rate curve. Specifically, 200 mg of DX-WPI-SP microcapsules were mixed with 10 mL of phosphate buffered solution with different pH and placed in a 25 mL glass bottle; 2 mL of the solution was taken out from the mixed solution at 5, 10, 30, 60, 90, and 120 min, respectively. The microcapsule suspension was centrifuged at 12,000 rpm for 2 min; 1 mL of the supernatant was taken out to determine the total selenium content through the atomic fluorescence spectrometer. The reference curve of the release rate of SP was calculated to describe the relationship between the SP concentration and the release time. All experiments were carried out in triplicate.

Mathematical models were utilized to better explain the release behavior of active substances. The zero order model and the first order model can be applied to describe the release behavior of the water-soluble components in water surroundings [34,35]. The Higuchi kinetic model is used to explain the water dissolution behavior of active substances in the solid and semi-solid matrix, and the Korsmeyer–Peppas kinetic model is usually applied to analyze the release mechanism of active substances from polymeric particle systems [36,37]. Herein, to evaluate the release mechanism of SP, multiple sustained-release kinetic models, as described above, were used to fit the release trend of DX-WPI-SP microcapsules based on the reported work [38]. The formulas are presented in the Appendix A.

### 2.6. Determination of In Vitro Release Properties of DX-WPI-SP Microcapsules

Gastric digestion and intestinal digestion were performed following the method reported by Gawlik-Dziki et al., with minor modifications [39]. Simulated gastric juice was prepared by adding 3.2 g of pepsin into 0.3 M of NaCl aqueous solution, with a pH value of 1.2, using 10% of HCl solution (*w*/*v*). Simulated intestinal juice was prepared by dissolving 0.35 g of trypsin and 0.3 g of bile in 35 mL of NaHCO_3_ aqueous solution (0.1 mol/L). The DX-WPI-SP microcapsules were incubated in the simulated gastric juice at 37 °C for 2 h and then introduced to the simulated intestinal juice for another 3 h. During the incubation, the amount of released SP in the supernatant was measured at different time intervals to calculate the release rate of SP. All experiments were conducted in triplicate.

### 2.7. Cell Culture and Vitality Assay In Vitro

The cytotoxicity of DX-WPI-SP microcapsules in the cell line Caco-2 (human colonic cancer cell line) was evaluated after digestion. Caco-2 was maintained in the DMEM medium containing 1% penicillin–streptomycin (PS) solution and 10% fetal bovine serum (FBS) at 37 °C in a humidified environment of 5% CO_2_ and 95% air [40]. Briefly, the Caco-2 cells were adhered at a density of 5.0 × 10^4^ cells/plate and cultured for 2–3 days. Then, the cells were sub-cultured in the same type of plate after trypsinization treatment.

A cell vitality assay was performed, referring to the reported method [41,42]; 100 μL of Caco-2 cell suspension at a density of 5 × 10^4^ cells/mL were adhered in a sterile 96-well plate to incubate for 24 h at 37 °C in a humidified environment of 5% CO_2_ and 95% air. Then, 10 μL of different concentrations of digested DX-WPI-SP microcapsule solution was added. Cell viability was conducted using the CCK-8 cell proliferation and cytotoxicity assay kit. Briefly, 10 μL of 2-(2-methoxy-4-nitrophenyl)-3-(4-nitrophenyl)-5-(2,4-disulfobenzene)-2H-tetrazole monosodium salt (WST) was added into a 96-well plate for incubation for 2 h. Then, the absorbance at 450 nm was recorded with a 96-well plate reader, and 6 parallel duplicate wells were set for each concentration. The cell viability calculation formula is presented in the Appendix A.

### 2.8. Statistical Analysis

All data were presented as mean ± standard deviation (SD). Duncan’s test was carried out to determine the significance of differences among the mean values (*p* < 0.05 was considered a significant level) via the SPSS Statistics 26 software program (SPSS, Inc., an IBM Company, Chicago, IL, USA).

## 3. Results and Discussion

### 3.1. Preparation of DX-WPI-SP Microcapsules

The preparation process of DX-WPI-SP microcapsules is shown in Figure 1. DX-WPI-SP microcapsules were fabricated via high-voltage electrospray using DX and WPI as shells, loaded with SP. The fitting of the electrohydraulic kinetic process depends on the properties of the polymer solution and the operation parameters [43]. The results of the optimization of DX-WPI-SP microcapsules are presented in Appendix A. The specific preparation process parameters and obtained diameters are shown in Table 1. The following parameters were summarized after the process optimization: 6% of DX (*w*/*v*), feeding rate (*Q*) 1.0 mL/h, applied voltage (*U*) 15 kV, and receiving distance (*H*) 15 cm. The gradual addition of WPI resulted in a mildly increasing diameter, and the decreasing loading rate of the microcapsules is shown in Figure 1. The DX-WPI-SP microcapsules obtained with these fabrication parameters have a spherical appearance and uniform size distribution (~40.06–~44.22 μm); the corresponding loading rate for SP ranged from ~46% to ~37% when the content of WPI (*w*/*v*) was 4–8%. The total Se content of the DX-WPI-SP microcapsules was calculated by referring to the standard curve, as shown in Appendix A.

The electrospraying solution ejected from the nozzle was subjected to numerous forces, including gravity and body forces (electrodynamic force, multidirectional stress, surface tangential stress, etc.). Under these comprehensive functions, the meniscus was destroyed and deformed to form a cone (Taylor cone). In addition, the electrospraying solution with free charge was accelerated and concentrated at the tip of the Taylor cone under the action of a potential difference and formed a jet, which was subsequently split into micro-scale droplets [18,44].

### 3.2. Structure and Characterization of DX-WPI-SP Microcapsules

Figure 2 shows the scanning electron microscope (SEM) images of DX microcapsules, DX-WPI microcapsules, and DX-WPI-SP microcapsules for surface morphology observation. Figure 2a shows that the pure DX microcapsules appear to have a smooth surface. However, Figure 2b shows that the DX-WPI microcapsules are irregularly shaped, with a rough surface. This is because the receiving solvent (anhydrous ethanol) grabs a lot of water within a short duration, resulting in the rough surface of the microcapsules and many voids after the ethanol reaches the microcapsules. Compared with the SEM image of DX-WPI microcapsules, there are more bumps and folds on the surface of the DX-WPI-SP microcapsules (Figure 2c).

FTIR was used to further identify the chemical bonding function and to characterize interactions between the functional groups of DX microcapsules, DX-WPI microcapsules, DX-WPI-SP microcapsules, WPI, and SP. The FTIR spectrum (Figure 3a) showed a characteristic peak at ~1640 cm^−1^, which is ascribed to the C=O stretching vibration within the amide I band. The characteristic peak at ~1590 cm^−1^ can be assigned to C-N stretching vibration in the amide II band [45]. The regions at 1000–1250 and 1300–1400 cm^−1^ indicate the existence of C-N and C-O bonds [46,47]. The FTIR spectrum of DX revealed a peak at ~3368 cm^−1^, which is due to the presence of -OH. After SP microencapsulation, a redshift from ~3300 to ~3286 cm^−1^ of this peak was noted for DX-WPI-SP microcapsules, which could be attributed to the enhancement of the hydrogen bond and a decreased number of free hydroxyl groups [48]. Therefore, the DX-WPI-SP microcapsules have a more compact structure. The peak at ~1271 cm^−1^ can be assigned to C-O, and the regions at ~917 and ~765 cm^−1^ are characteristic peaks from dextran [49]. In addition, all characteristic peaks of SP, DX, and WPI appeared in the FTIR spectrum of the DX-WPI-SP microcapsules, which provided good evidence that the SP was successfully loaded onto the DX-WPI microcapsules. It was notable that no new peaks appeared in the FTIR spectrum of the DX-WPI-SP microcapsules, suggesting that no new chemical bond was formed between DX-WPI and SP, and DX-WPI was used as the SP carrier without significant change to its inherent properties.

X-ray diffraction (XRD) was used to analyze the crystal phase of the substances. As shown in Figure 3b, the SP source had characteristic peaks at ~11.7°, ~20.8°, ~29.1°, ~31.2°, and ~33.4°, indicating the SP has a crystalline nature. XRD patterns of the DX microcapsules showed that the peak at 2θ = ~18.9° was broad and round, exhibiting that the DX microcapsules were amorphous [50]. After DX microcapsules were complexed with WPI, a new peak appeared at 2θ = ~8.9°, which was attributed to the addition of WPI. The XRD pattern of the DX-WPI-SP microcapsules revealed that this compound was mainly amorphous. All typical bands of the DX-WPI microcapsules remained within the XRD pattern of the DX-WPI-SP microcapsules.

To investigate the thermal stability and degradation process, the thermal stability of DX microcapsules, DX-WPI microcapsules, and DX-WPI-SP microcapsules, as well as SP, was evaluated with TGA under a nitrogen atmosphere. The TGA and DTG thermograms are shown in Figure 3c,d. In Figure 3c, all substances exhibit a typical two-step thermal degradation process in their TGA thermograms. The first stage is with a minor weight loss, which is attributed to the loss of moisture. The second stage shows a heavy weight loss as a result of the thermal decomposition of the chemical compounds [51]. The degradation rate of the DX-WPI microcapsules was lower than that of the DX microcapsules, which maybe due to the improved heat resistance of the backbone network in the compound because of the enhancement of interactions between proteins and polysaccharides, such as hydrophobic interactions and hydrogen bonding functions. Figure 3d shows that free SP has a clear heat degradation process from ~120 to ~300 °C. However, after microencapsulation, the degradation rate is significantly slowed down, which is coincident with the fact that the wall material can protect the SP against thermal degradation and improve its thermal stability. Researchers have also found that the thermal stability of squalene was improved after encapsulation treatment by polysaccharide and protein comparing to that of pure squalene [52].

### 3.3. Antioxidant Activity Evaluation of DX-WPI-SP Microcapsules

The DPPH free radical scavenging test and the ABTS free radical scavenging test were used to evaluate the antioxidant properties of substances. Figure 4a,b show the DPPH and ABTS free radical scavenging rates of SP under different mass concentrations. SP exhibited increasing DPPH and ABTS free radical scavenging activities with the gradual introduction of SP. Figure 4c,d show the DPPH and ABTS radical scavenging activity comparisons of DX microcapsules, DX-WPI microcapsules, DX-WPI-SP microcapsules, and SP, respectively. The antioxidant activity of DX-WPI-SP microcapsules was only ~67.7% of that of free SP. It was noteworthy that the antioxidant activity of DX-WPI-SP microcapsules was significantly higher than that of both DX microcapsules and DX-WPI microcapsules. The complexation of SP could enhance the antioxidant activity of the DX-WPI microcapsules, and the DX-WPI-SP microcapsules had excellent antioxidant capacity.

### 3.4. Controlled-Release Properties Evaluation Results

Figure 5a shows the release behavior of DX-WPI-SP microcapsules under different pH conditions (pH 2.0, pH 7.0, and pH 12.0). The SP was rapidly released from the DX-WPI-SP microcapsules within 10 min, with a release rate of 70–80%, and then the release rate increased slowly and held steady. The release rate of the SP from the DX-WPI-SP microcapsules was faster in the initial release stage, resulting in a large accumulated SP content. As the infiltration process was delayed, the release rate of SP gradually slowed down and held steady at about 70–85%. The SP was more easily released from the encapsulated microcapsules at lower pH values, where the DX-WPI wall material was more easily collapsed. Similar phenomena occurred in alginate-whey protein isolate microcapsules [53]. The strong basic environment affected non-covalent interactions within the WPI, including van der Waals force, hydrophobic interaction, hydrogen bonding function, etc. [54]. In strong basic surroundings, the DX-WPI wall was harder to disintegrate from DX-WPI-SP compared to acidic environments, which was attributed to the excessive exposure of the hydrophobic groups of WPI [55]. The enhancement of hydrophobic interactions decreased the solubility of WPI and led to its aggregation in water environments [56,57]. Thus, the release rate of SP from DX-WPI-SP microcapsules was lower at the higher pH values, and the DX-WPI-SP microcapsules could also show controllable release properties for SP.

The release mechanism model for active substance release can essentially reveal the influence of release dynamics and release architecture variables [38]. Diffusion, swelling, and erosion are the main release processes that affect the control and release behaviors of the active ingredients. The higher the local activated material concentration, the faster the diffusion efficiency of SP from the DX-WPI-SP microcapsules. Similar results were reported by Ge et al. and Wu et al. [58,59]. The kinetic parameters are shown in Appendix A for the release process of SP from DX-WPI-SP microcapsules in different pH values, and the results of the fitting of release kinetics are presented in the Appendix A.

The release performance of SP from the DX-WPI-SP microcapsules was monitored in the simulated gastrointestinal tract (Figure 1). As shown in Figure 5b, the SP was rapidly released within 120 min, with a release rate of ~90%, which is consistent with the release behavior of SP from the DX-WPI-SP microcapsules under acidic conditions (pH 2.0) (Figure 5a). The conformational change of WPI under acidic conditions can make pepsin more accessible to targets with amino acid residues, and the enzymatic hydrolysis of WPI can increase the release of SP. Nongonierma et al. gathered the similar result that the release rate of bioactive peptides was enhanced via enzymatic hydrolysis of milk proteins [60].

### 3.5. Cytotoxicity Study of Digested DX-WPI-SP Microcapsules

The cellular cytotoxicity of DX-WPI-SP microcapsules by the treatment of a digestive solution for Caco-2 cells was determined by the CCK-8 kit. As shown in Table 2, the Se content showed a negligible influence on the cellular cytotoxicity of Caco-2 cells, where the cell viability was still over 90% even when the Se content was up to ~46 μg/mL. Therefore, it is implied that DX-WPI-SP microcapsules have quite low toxicity and great biocompatibility, and they also show potential as selenium supplements and are an excellent prospect in the food field.

## 4. Conclusions

In summary, a facile strategy of a selenium-enriched peptide (SP, selenopeptide) carrier has been reported for the protection and delivery of SP. Dextran–whey protein isolation–selenopeptide (DX-WPI-SP) microcapsules were fabricated via high-voltage electrospraying technology. This work has optimized the process parameters to obtain microcapsules with suitable particle size and high loading efficiency. The optimized process parameters include the following: concentration of dextran (DX) 6%, feeding rate 1.0 mL/h, applied voltage 15 kV, and receiving distance 15 cm. When the concentration of whey protein isolation (WPI) was 4–8%, the corresponding diameter of the DX-WPI-SP microcapsules was below 45 μm, and the loading rate for SP was in the range of ~46% to ~37%. Compared with free SP, the thermal stability of microencapsulated SP (DX-WPI-SP) was significantly improved due to the protection function of the wall materials of DX-WPI. The release properties of SP from the DX-WPI-SP microcapsules were monitored under different pH values, where the DX-WPI-SP microcapsules exhibited a controllable release effect on SP. The SP release dynamic from the microcapsules was nearly in accordance with the first order model (pH 2.0) and the Korsmeyer–Peppas model (pH 7.0 and pH 12.0). The results of in vitro simulation experiments show that the DX-WPI-SP microcapsules have excellent gastrointestinal dissolution characteristics and controllable release properties for SP. Herein, this work provides a valuable tool to fabricate and encapsulate selenium-enriched materials and also provides a vehicle for the controllable release of SP in vitro. The constructed system has outstanding biocompatibility and negligible cellular cytotoxicity, exhibiting prospects and potential in the food industry.

## Figures and Tables

**Figure 1 foods-12-01008-f001:**
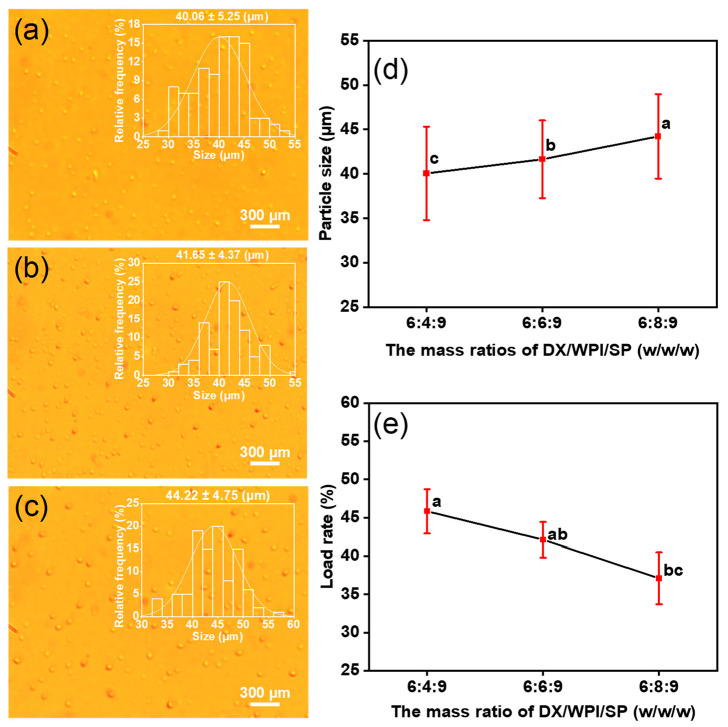
(**a**–**c**) Optical images and particle size distribution of DX-WPI-SP microcapsules with different DX/WPI/SP mass ratios (a—6:4:9, b—6:6:9, c—6:8:9). (**d**) The average particle size and (**e**) the average loading rate of these DX-WPI-SP microcapsules against different mass ratios of DX/WPI/SP (6:4:9, 6:6:9, 6:8:9, *w*/*w*/*w*). Different letters represent significant differences between groups by Duncan’s test (*p* < 0.05).

**Figure 2 foods-12-01008-f002:**
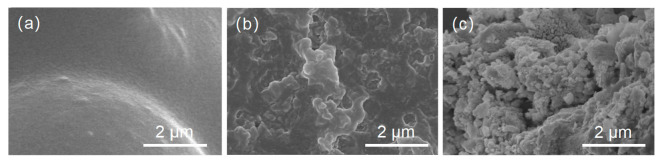
SEM images of (**a**) DX microcapsules, (**b**) DX-WPI microcapsules, and (**c**) DX-WPI-SP microcapsules.

**Figure 3 foods-12-01008-f003:**
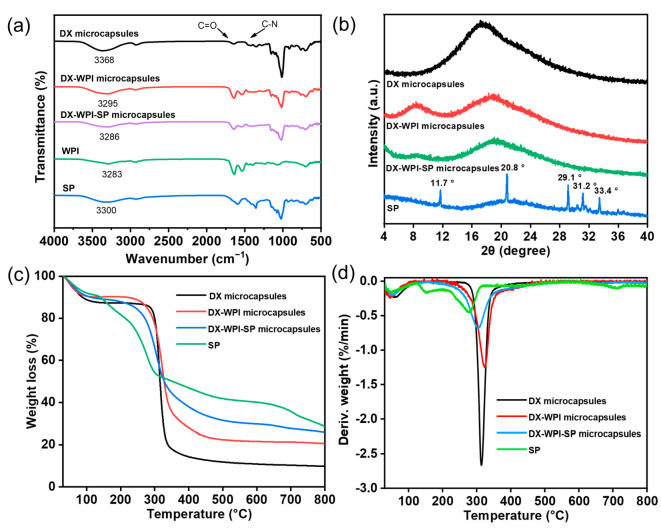
(**a**) FTIR curves, (**b**) XRD patterns, (**c**) TGA curves, and (**d**) DTG curves of DX microcapsules, DX-WPI microcapsules, DX-WPI-SP microcapsules, WPI, and SP.

**Figure 4 foods-12-01008-f004:**
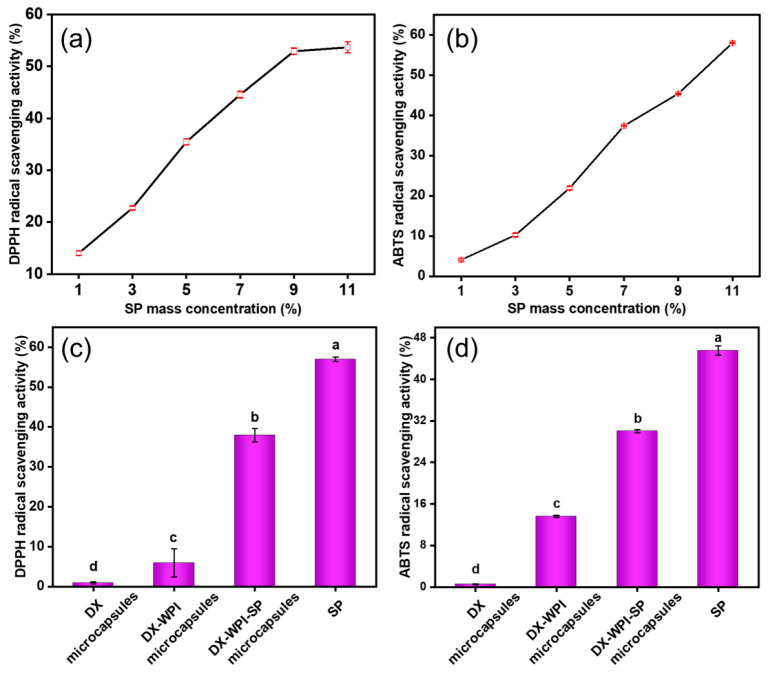
(**a**) DPPH and (**b**) ABTS radical scavenging activity against various SP concentrations. (**c**) DPPH and (**d**) ABTS radical scavenging activity comparisons of DX microcapsules, DX-WPI microcapsules, DX-WPI-SP microcapsules, and SP (all of them were 9%, *w*/*v*). Different letters represent significant differences between columns by Duncan’s test (*p* < 0.05).

**Figure 5 foods-12-01008-f005:**
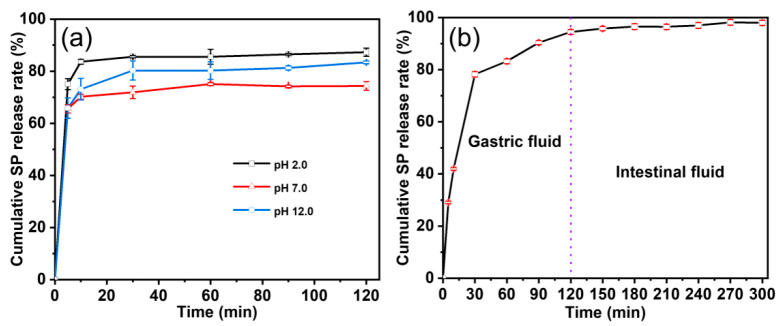
The cumulative release curves of SP released from DX-WPI-SP microcapsules under (**a**) the different pH values (pH 2.0, pH 7.0, and pH 12.0) and (**b**) the simulated gastric and intestinal digestion surroundings.

**Scheme 1 foods-12-01008-sch001:**
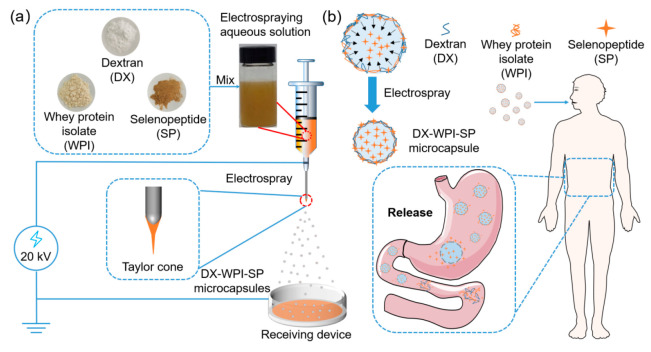
Schematic illustration of (**a**) the preparation process of DX-WPI-SP microcapsules and (**b**) the delivery system of DX-WPI-SP microcapsules in the human gastrointestinal tract.

**Table 1 foods-12-01008-t001:** Operation parameters for various microcapsules via electrospraying and the diameter of as-prepared microcapsules.

Microcapsule	DX (g)	WPI (g)	SP (g)	Voltage (kV)	Distance (cm)	Feeding Rate (mL/h)	Diameter (μm)
DX microcapsules	0.6	-	-	15	15	1.0	26.47 ± 3.50
DX-WPI microcapsules	0.6	0.2	-	15	15	1.0	30.31 ± 2.91
	0.6	0.4	-	15	15	1.0	30.75 ± 3.64
	0.6	0.6	-	15	15	1.0	33.42 ± 3.09
	0.6	0.8	-	15	15	1.0	34.38 ± 4.07
	0.6	1.0	-	15	15	1.0	35.83 ± 3.64
DX-WPI-SP microcapsules	0.6	0.4	0.9	15	15	1.0	40.06 ± 5.25
	0.6	0.6	0.9	15	15	1.0	41.65 ± 4.37
	0.6	0.8	0.9	15	15	1.0	44.22 ± 4.75

**Table 2 foods-12-01008-t002:** Cytotoxicity of digestive solutions at different dilutions.

Digestive Solution at Different Dilutions	Total Selenium Content (μg/mL)	Cell Viability (%)	*p* Value ^a^
Control	-	100	-
Digestive solution	45.72 ± 2.98	97.21 ± 2.57 ^a^	0.9127 ^ns^
Digestive solution-2	22.86 ± 1.49	98.33 ± 2.90 ^a^	0.9925 ^ns^
Digestive solution-10	4.57 ± 0.30	95.94 ± 4.00 ^a^	0.6711 ^ns^
Digestive solution-20	2.29 ± 0.15	97.94 ± 5.22 ^a^	0.9783 ^ns^
Digestive solution-50	1.14 ± 0.06	97.58 ± 0.66 ^a^	0.9533 ^ns^
Digestive solution-100	0.57 ± 0.03	96.69 ± 2.93 ^a^	0.8298 ^ns^

^a^ Duncan’s significance test. ^ns^
*p* > 0.05 is considered as not significant compared with the control group. Digestive solution-X (2, 10, 20, 50, and 100): X represents the diluted times of the digestive solution with PBS buffer solution.

## Data Availability

The data presented in this study are available upon request from the corresponding author. The data are not publicly available because of privacy issues.

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
