# Peer review of "Electrospray-Assisted Fabrication of Dextran–Whey Protein Isolation Microcapsules for the Encapsulation of Selenium-Enriched Peptide"

_foods, 2023, doi:10.3390/foods12051008_

Round 1
Reviewer 1 Report
Overall, I found the manuscript titled "Electrospray-Assisted Fabrication of Dextran-Whey Protein Isolation Microcapsules for the Encapsulation of Selenium-Enriched Peptide: Structural and Functional Characteristics." to be well-written and well-organized, with clear and concise descriptions of the research methods and results.
One area where the manuscript could be improved is in the readability of the figures. In particular, Figure 1. could be improved in this regard. However, this is a minor issue and does not significantly impact the overall quality and value of the manuscript.
The authors present an optimized method for fabricating microcapsules using electrospray technology and demonstrate its effectiveness for encapsulating selenium-enriched peptides. The structural and functional characteristics of the resulting microcapsules are thoroughly analyzed and discussed, providing valuable insights into the potential applications of this technology.
I believe it will be of interest to a wide audience of scientists and researchers. I highly recommend this manuscript, with the suggestion that the readability of the figures is improved in future versions.
Author Response
Authors’ response
Dear editor and referees,
First of all, we would like to express our sincere thanks for your kind consideration and extremely valuable comments for the manuscript entitled “Electrospray-Assisted Fabrication of Dextran-Whey Protein Isolation Microcapsules for the Encapsulation of Selenium-Enriched Peptide: Structural and Functional Characteristics” (Manuscript ID: foods-2112689). We have made efforts to follow the referees’ suggestions and revise the manuscript according to the comments. The specific modifications are shown below (changes in the manuscript are highlighted in red). Attached are the itemized responses to the referees’ reports, which also summarize the changes made in the manuscript. We hope you will find these responses and changes satisfactory and will meet the requirement for approval of the manuscript.
If you have any other questions, please contact us: caijievip@whpu.edu.cn
Sincerely yours,
A/Prof. Jie CAI
Feb. 2, 2023
Comments and Suggestions for Authors
Comment 1: Overall, I found the manuscript titled "Electrospray-Assisted Fabrication of Dextran-Whey Protein Isolation Microcapsules for the Encapsulation of Selenium-Enriched Peptide: Structural and Functional Characteristics." to be well-written and well-organized, with clear and concise descriptions of the research methods and results.
Response: Thank you for your encouraging comment.
Comment 2: One area where the manuscript could be improved is in the readability of the figures. In particular, Figure 1. could be improved in this regard. However, this is a minor issue and does not significantly impact the overall quality and value of the manuscript.
Response: Thanks for your professional suggestion. We have improved the figure following your suggestion as shown in Figure 1 in the revised manuscript.
Comment 3: The authors present an optimized method for fabricating microcapsules using electrospray technology and demonstrate its effectiveness for encapsulating selenium-enriched peptides. The structural and functional characteristics of the resulting microcapsules are thoroughly analyzed and discussed, providing valuable insights into the potential applications of this technology.
Response: Thank you for the recognition for our work.
Comment 4: I believe it will be of interest to a wide audience of scientists and researchers. I highly recommend this manuscript, with the suggestion that the readability of the figures is improved in future versions.
Response: Thank you very much for your recommendation and suggestion. We have improved the Figure 1 in the revised manuscript and hope that the modification will meet with your satisfaction and requirement.

Reviewer 2 Report
The article reported synthesis of the dextran-whey protein isolation microcapsules to encapsulate peptide with selenium as supplement. The article was well written, with minor revisions required as follows.
The first paragraph in introduction is too long, it needs to be separated. The second paragraph can be started with the word of “Microencapsulation...” in Line 51.
Line 105, Q is a feeding rate, while in Abstract Q is a pushing speed. It is better to use the same term and to make consistent.
All the abbreviations in line 211-217 needs to be removed. They are already defined in the Introduction.
In 226, the author used “~” for the loading rate, why was it used? why did not use standar deviation, instead of using the mark. It also used for others, such as characteristic peak, etc.
Table 1 shows diameter in unit “nm”, but in the Figure 1, “μm” was used. Please recheck!
Author Response
Authors’ response
Dear editor and referees,
First of all, we would like to express our sincere thanks for your kind consideration and extremely valuable comments for the manuscript entitled “Electrospray-Assisted Fabrication of Dextran-Whey Protein Isolation Microcapsules for the Encapsulation of Selenium-Enriched Peptide: Structural and Functional Characteristics” (Manuscript ID: foods-2112689). We have made efforts to follow the referees’ suggestions and revise the manuscript according to the comments. The specific modifications are shown below (changes in the manuscript are highlighted in red). Attached are the itemized responses to the referees’ reports, which also summarize the changes made to the manuscript. We hope you will find these responses and changes satisfactory.
If you have any other questions, please contact us: caijievip@whpu.edu.cn
Sincerely yours,
A/Prof. Jie CAI
Feb. 2, 2023
Comments and Suggestions for Authors
Comment 1: The article reported synthesis of the dextran-whey protein isolation microcapsules to encapsulate peptide with selenium as supplement. The article was well written, with minor revisions required as follows.
Response: Thanks so much for your suggestions. We have revised the manuscript according to your opinions, and sincerely hope that the modification will meet with your satisfaction.
Comment 2: The first paragraph in introduction is too long, it needs to be separated. The second paragraph can be started with the word of “Microencapsulation...” in Line 51.
Response: Thank you for your suggestion. The first paragraph was separated to three paragraphs as shown in the revised manuscript. Meanwhile, “Microencapsulation...” was used in Line 52 as shown in the revised manuscript.
Comment 3: Line 105, Q is a feeding rate, while in Abstract Q is a pushing speed. It is better to use the same term and to make consistent.
Response: Thanks a lot for this great advice. We have uniformly defined “Q” as feeding rate, and the pushing speed was changed to feeding rate in the “Abstract” in Line 23 as shown in the revised manuscript following your suggestion.
Comment 4: All the abbreviations in line 211-217 needs to be removed. They are already defined in the Introduction.
Response: Thanks very much, these abbreviations have been removed.
Comment 5: In 226, the author used “~” for the loading rate, why was it used? why did not use standard deviation, instead of using the mark. It also used for others, such as characteristic peak, etc.
Response: “~”means “about”, “around” or “approximately”. “~” is not the meaning of loading rate in Line 221-222 as shown in the revised manuscript.
Comment 6: Table 1 shows diameter in unit “nm”, but in the Figure 1, “μm” was used. Please recheck!
Response: We are extremely sorry for this mistake. We have revised “nm” to “μm” in Table 1 as shown in the revised manuscript.
We thank you again, and hope that the modifications for the manuscript according to your suggestions will meet the requirement of the journal for approval.

Reviewer 3 Report
After reviewing a manuscript, I suggest following corrections:
1. Line 62: Poly(D, L-lactide-coglycolide), lower case- poly….
2. Lines 77, 79, 129, 138, 147, etc: Citation of references-e.g. M. Costamagna et al. reported that… or Costamagna et al. reported…..
3. Lines 184-186: Which method was used to measure the total amount of SP in supernatant? Was the total selenium content determined (as described in supplementary material)? Or some other method for measuring peptide-Se complex was used. Is it pure peptide-Se complex or its semi purified powder?
4. Is there any additional information (chemical characterization) of selenium-enriched peptide powder from Cardamine violifolia, which was used in this study e. g. degree of purification- it is important for antioxidant potential measurements. Other ingredients of the investigated powder enriched with Se-peptide e.g. polyphenols or polyphenol complex with proteins/peptides may have impact on the antioxidant potential. Did the type of interaction among Se and peptide investigate, e.g. linkage type, formation of complex….?
5. Line 389: …. release properties for SP. Is SP purified or semi purified powder enriched with Se-peptide.
6. Figure 3a and lines 264-265: The FTIR spectrum (Figure 3a) of WPI showed a characteristic peak at ~1640 cm−1, which was ascribed to C=O bending vibration within the amide I band. The characteristic peak at ~1590 cm-1 belongs to the amide II band, which was mainly due to the bending vibration of C=N [45].
There are no comments and explanations of SP spectrum, e.g. absorption bands at region 1000-1250 and 1300–1400 cm−1. Can protein characteristic absorptions at 1590-1640 cm−1 overlapped with absorption of some other compounds (if SP was not completely purified).
Author Response
Authors’ response
Dear editor and referees,
First of all, we would like to express our sincere thanks for your kind consideration and extremely valuable comments for the manuscript entitled “Electrospray-Assisted Fabrication of Dextran-Whey Protein Isolation Microcapsules for the Encapsulation of Selenium-Enriched Peptide: Structural and Functional Characteristics” (Manuscript ID: foods-2112689). We have made efforts to follow the referees’ suggestions and revise the manuscript according to the comments. The specific modifications are shown below (changes to the manuscript are highlighted in red). Attached are the itemized responses to the referees’ reports, which also summarize the changes made to the manuscript. We hope you will find these responses and changes satisfactory.
If you have any other questions, please contact us: caijievip@whpu.edu.cn
Sincerely yours,
A/Prof. Jie CAI
Feb. 2, 2023
Comments and Suggestions for Authors
After reviewing a manuscript, I suggest following corrections:
Comment 1: Line 62: Poly(D, L-lactide-coglycolide), lower case- poly….
Response: Thanks for your suggestion, we have used lowercase for “Poly” to “poly” in Line 61 as shown in the revised manuscript following your suggestion.
Comment 2: Lines 77, 79, 129, 138, 147, etc: Citation of references-e.g. M. Costamagna et al. reported that… or Costamagna et al. reported…..
Response: Thanks so much for your valuable comment. We have carefully checked these references again; they have been correctly cited and used.
Comment 3: Lines 184-186: Which method was used to measure the total amount of SP in supernatant? Was the total selenium content determined (as described in supplementary material)? Or some other method for measuring peptide-Se complex was used. Is it pure peptide-Se complex or its semi purified powder?
Response: Thanks a lot for your kind and valuable comment. Selenium (Se)-enriched peptides were isolated from Cardamine violifolia by enzymatic hydrolysis and ultrafiltration. The total selenium content was determined on a LC-AFS6500 atomic fluorescence spectrometer (Haiguang Instrument, China), as shown in the Supplementary Materials. And the total Se content of SP was 1143.65 ± 64.27 μg/g. And we have provided this statement in Materials and reagents as shown in the revised Supplementary Materials. Moreover, SP is a kind of semi purified powder.
Comment 4: Is there any additional information (chemical characterization) of selenium-enriched peptide powder from Cardamine violifolia, which was used in this study e. g. degree of purification- it is important for antioxidant potential measurements. Other ingredients of the investigated powder enriched with Se-peptide e.g. polyphenols or polyphenol complex with proteins/peptides may have impact on the antioxidant potential. Did the type of interaction among Se and peptide investigate, e.g. linkage type, formation of complex….?
Response: Thank you very much for your kind comment.Selenium-enriched peptide (SP) is a kind of semi purified powder and it is a quite complex composite. We have provided the total Se content (1143.65 ± 64.27 μg/g) of SP powder from Cardamine violifolia in Materials and reagents as shown in the revised Supplementary Materials. According to the previous studies, selenium-enriched peptide powder from Cardamine violifolia has been demonstrated that SP supplementation could ameliorate D-galactose (D-gal) induced metabolism disorder, memory dysfunction, and neuron damage in the hippocampus1. Additionally, SP supplementation can prevent obesity and metabolic disorders caused by high-fat diet, probably by ameliorating oxidative stress and inflammation, regulating metabolic genes, and modulating the gut microbiota compositions2.
Comment 5: Line 389: …. release properties for SP. Is SP purified or semi purified powder enriched with Se-peptide.
Response: Thank you very much for your valuable comment. Selenium-enriched peptide (SP) used in our study is a kind of semi purified powder. And we have provided this statement in Materials and reagents of the revised Supplementary Materials.
Comment 6: Figure 3a and lines 264-265: The FTIR spectrum (Figure 3a) of WPI showed a characteristic peak at ~1640 cm−1, which was ascribed to C=O bending vibration within the amide I band. The characteristic peak at ~1590 cm-1 belongs to the amide II band, which was mainly due to the bending vibration of C=N [45].
There are no comments and explanations of SP spectrum, e.g. absorption bands at region 1000-1250 and 1300–1400 cm−1. Can protein characteristic absorptions at 1590-1640 cm−1 overlapped with absorption of some other compounds (if SP was not completely purified).
Response: Thank you for your comment. We have supplemented the FTIR analysis for SP, in which the regions at 1000-1250 cm−1 and 1300-1400 cm−1 indicate the existence of C-N and C-O bonds3, 4. And the characteristic peaks at ~1590 cm−1 and ~1640 cm−1 mainly corresponding to the stretching vibrations of C-N and C=O, respectively. It is worth mentioning that SP is a kind of semi purified powder and it is a quite complex composite rather than a pure chemical. This information is also provided in Line 259-262 of the revised manuscript.
References:
- Yu, T.; Guo, J.; Zhu, S.; Zhang, X.; Zhu, Z. Z.; Cheng, S.; Cong, X. Protective effects of selenium-enriched peptides from Cardamine violifolia on d-galactose-induced brain aging by alleviating oxidative stress, neuroinflammation, and neuron apoptosis. Journal of Functional Foods 2020, 75, 104277.
- Yu, T.; Guo, J.; Zhu, S.; Li, M.; Zhu, Z.; Cheng, S.; Wang, S.; Sun, Y.; Cong, X. Protective effects of selenium-enriched peptides from Cardamine violifolia against high-fat diet induced obesity and its associated metabolic disorders in mice. RSC Advances 2020, 10, 31411-31424.
- He, J.; He, Y.; Zhuang, J.; Zhang, H.; Lei, B.; Liu, Y.; Luminescence properties of Eu3+/CDs/PVA composite applied in light conversion film. Optical Materials 2016, 62, 458-464.
- Li, W.; Wu, S.; Xu, X.; Zhuang, J.; Zhang, H.; Zhang, X.; Hu, C.; Lei, B.; Kaminski, C.F.; Liu, Y. Carbon dot-silica nanoparticle composites for ultralong lifetime phosphorescence imaging in tissue and cells at room temperature. Chemistry Materials 2019, 31, 9887-9894,

Reviewer 4 Report
The manuscript focuses on structural and functional characterizations of selenium-enriched peptide-loaded into dextran-whey protein microcapsules. Overall the manuscript's content is well organized. however, it has some deficiencies. detailed comments are as follows.
- The title should be summarized.
- In Supplementary Materials, there wasn’t references for characterizations of microparticles. Please mention the reference below for charactrization section.
https://doi.org/10.1016/j.fochx.2021.100202
https://doi.org/10.1016/j.ijbiomac.2021.11.155
- Figure 1: Please change the background color of figure 1 a-c.
- Figure 4 caption. Groups should be changed to columns.
Author Response
Authors’ response
Dear editor and referees,
First of all, we would like to express our sincere thanks for your kind consideration and extremely valuable comments for the manuscript entitled “Electrospray-Assisted Fabrication of Dextran-Whey Protein Isolation Microcapsules for the Encapsulation of Selenium-Enriched Peptide: Structural and Functional Characteristics” (Manuscript ID: foods-2112689). We are pleased to follow the referees’ suggestions and revise the manuscript according to the comments. The specific modifications are shown below (changes to the manuscript are highlighted in red). Attached are the itemized responses to the referees’ reports, which also summarize the changes made to the manuscript. We hope you will find these responses and changes satisfactory.
If you have any other questions, please contact us: caijievip@whpu.edu.cn
Sincerely yours,
A/Prof. Jie CAI
Feb. 2, 2023
Comments and Suggestions for Authors
Comment 1: The manuscript focuses on structural and functional characterizations of selenium-enriched peptide-loaded into dextran-whey protein microcapsules. Overall, the manuscript's content is well organized. however, it has some deficiencies. detailed comments are as follows.
Response: Thanks so much for your professional comments. We are so glad to upgrade this manuscript based on your suggestions, and sincerely hope you can find this revision satisfactory.
Comment 2: The title should be summarized.
Response: Thanks a lot. We renamed the title of this manuscript as shown in the revised manuscript and supplementary materials.
Comment 3: In Supplementary Materials, there wasn’t references for characterizations of microparticles. Please mention the reference below for characterization section.
https://doi.org/10.1016/j.fochx.2021.100202
https://doi.org/10.1016/j.ijbiomac.2021.11.155
Response: Thanks for your advice. We supplemented these two references for the characterizations of microparticles as shown in the revised Supplementary Materials.
Comment 4: Figure 1: Please change the background color of figure 1 a-c.
Response: Thanks so much. Figure 1a-c represent the optical images, which were captured under the self-installed light source. Thus, we improved the figure resolution to make the figure clearer.
Comment 5: Figure 4 caption. Groups should be changed to columns.
Response: Thanks for your suggestion. We changed the “groups” to “columns” as shown in the Figure 4 caption.

Round 2
Reviewer 3 Report
A suggestion for further research to investigate whether other components of the semi-purified selenium-peptide-enriched powder affect the antioxidant potential.
Reviewer 4 Report
The revisions have been completed by authors.